# Analysis of Demand and Structural Changes in China's Water Resources over the Next 30 Years

**Qiang Yan [1,2], Yan Zhang [1,2] and Qingwei Wang [3,***

[1]  Ministry of Natural Resources Key Laboratory of Metallogeny and Mineral Assessment, Institute of Mineral Resources, Chinese Academy of Geological Sciences, Beijing 100037, China

[2]  Research Center for Strategy of Global Mineral Resources, Chinese Academy of Geological Sciences, Beijing 100037, China

[3]  College of Geosciences and Engineering, North China University of Water Resources and Electric Power, Zhengzhou 450045, China

[*]  Correspondence: qingweiwang@126.com

**Abstract:** The medium and long-term demand prediction for water resources is critical to the country's macro-allocation of water resources and the formulation of a long-term regional economic development strategy. However, the official prediction report of China's water resources was created ten years ago, and its results are far from accurate in practice making it difficult to direct China's water resource planning in the coming ten years. By introducing the demand forecasting method for the first time, this study built a mathematical prediction model that integrates "data constraint + mathematical prediction + actuarial prediction". The conclusions are as follows: (1) The new method developed in this research is more accurate than previous prediction methods and has a higher degree of matching with actual data. Despite the problem of trend extrapolation, this model can accurately understand the long-term trend. However, the amount of data must be calculated. (2) China's consumption of total water, agricultural water, industrial water, domestic water, and ecological water will predict to be 552.9 billion $m^3$, 319.1 billion $m^3$, 95.7 billion $m^3$, 95.5 billion $m^3$ and 42.6 billion $m^3$ in 2030, respectively, as well as 504.7 billion $m^3$, 281.1 billion $m^3$, 61.4 billion $m^3$, 101.4 billion $m^3$, and 60.8 billion $m^3$ in 2050, respectively, indicating obvious structural changes. (3) China's industrial structure evolves frequently; therefore, water resource forecasting must account for changes in industrial structure as well as geographical variances. In North China, agricultural water demand will fall, but ecological water investment will rise. The planning and distribution of the nation's water resources will significantly be influenced theoretically and practically by the development of this methodology study. This method is universal and can be used for long-term prediction of future water demand in other countries.

**Keywords:** water resources; demand forecasting; model building; coupled analysis





## 1. Introduction

Water resources are irreplaceable natural resources in human social activities and play a vital role in economic and social development [1,2]. However, due to the lack of emphasis on water resources in China in the past period, the excessive development and utilization of water resources for economic development will eventually lead to water shortages, and the contradiction between supply and demand is prominent [3,4]. On the other hand, the medium and long-term demand forecast of water resources has great practical guiding significance for the country to carry out the macro-allocation of water resources and formulate long-term planning for regional economic development [5,6]. How to alleviate the contradiction between water supply and demand by predicting the demand for water resources has become an important task of water resources planning in various countries and regions. Since the 1960s, Western countries such as the United States have implemented domestic water demand forecasting, which provides guidance for the development and

utilization of water resources [7–10]. Based on historical water consumption data, the United States has carried out a forecast and analysis of the actual water demand in the future, taking into consideration the water-saving situation half of the time. The actual water consumption data in the United States showed that the actual water demand was consistent with the predicted results [8,9]. Following from the above, in the late 1970s, Japan, the United Kingdom, and other Western developed countries conducted water resource forecasting and planning studies [10]. In order to solve the global water problem collaboratively, the United Nations releases the "United Nations World Water Development Report" every year, which includes practical recommendations for prospective or present water emergencies. Water scarcity and waste are issues that China and many other nations across the world are dealing with [11–13]. Therefore, China has also performed three studies on mid- to long-term forecasting, planning, and assessment of water resources, resulting in three reports: "2001 Data Report Collection [14]", "2010 Strategic Research [15]", and "2010 Planning [16]". However, due to society's fast growth, even the recent water resource projection study is approximately twelve years old, and its anticipated findings have departed from actual water use. The following are the specific reasons: (1) The entire population forecast is too high. According to the State Council's national population growth plan (2016–2030), China's population will reach 1.42 billion by 2020 and peak at 1.45 billion around 2030. Clearly, these studies employed an overestimation of China's projected population (Table 1) [14–16], compromising the accuracy of future water demand forecasting; the water consumption projection is on the high side. (2) The forecasted rate of urbanization is too slow. Because there is a significant disparity in water use between urban and rural inhabitants, forecasting urbanization will also affect forecasting future water demand. In 2016, China's urbanization rate had reached 57.40%, surpassing 2030's level projected in 2001 (Table 1) [14–16]. Therefore, in recent years, the rate of urbanization in China has clearly surpassed the forecasts utilized in these studies. (3) The mismatch between the predicted and actual industrial structure. The amount of water used varies substantially amongst industries. In 2016, the share of the economy's secondary sector was less than 40%, while the proportion of the economy's tertiary sector was greater than 51.6%. The proportion of primary industry and the secondary sector of the economy industry will continue to fall in the future, while the proportion of the tertiary sector of the economy industry will rise. The anticipated statistics for China's future industrial structure of the three reports deviate significantly from the actual reality (Table 1) [14–16]. (4) Lack of attention to ecological water. It is critical to rationally allocate our country's ecological water for the fulfillment of sustainable development of our ecological environment. Although there are some references to previous ecological water use, it is a small part, and the forecast method of future ecological water use is not flawless; most are approximate predictions with low accuracy.

It is clearly impractical to use it to guide China's water resource planning for the next 10 years [5], and it must be changed in light of societal evolution. In order to forecast the water consumption of agriculture, industry, life, and ecology in China in 2050, this paper proposes a mathematical prediction model using the grey system, regression equation, etc., and then combines it with scenario analysis, analogy method, and reference to economic and social data for "actual revision" before creating a water resources demand forecasting model that integrates "data constraint + mathematical prediction + actuarial prediction". The model combines qualitative and quantitative methods, and the results are mutually and completely validated. By overcoming the disadvantage of trend extrapolation, this study is able to comprehend the long-term trend accurately. Its' essential notion is the creation of accurate and reliable large data, and it synthesizes many diverse forecasting methodologies and mathematical models. Despite the workload, maximum accuracy can be guaranteed. In the qualitative prediction, we collected and collated long-scale and multi-sample water resources and economic data from more than 20 Western developed countries and analyzed the causes of change in water use, the general law of the relationship between water resources consumption and economic development, population, urbanization rate, planting

structure, and industrial structure. At the same time, according to China's basic national conditions, important strategic planning, policy, and other future water use scenario analyses to achieve the overall trend of demand are used to grasp the trend. In the quantitative prediction, the Grey system model, quota method, analogy analysis, regression analysis, scenario analysis, and actual data constraint are used to forecast.

**Table 1.** Comparison of the predicted data of different research results.

| Items | «Comprehensive Report of Strategy on Water Resources for China's Sustainable Development» [14] | | «Research on Water Resources and Sustainable Development Strategy in China» [15] | | «Comprehensive National Water Resources Planning (2010–2030)» [16] | | Actual Data | |
|---|---|---|---|---|---|---|---|---|
| | **2010** | **2030** | **2020** | **2030** | **2020** | **2030** | **2010** | **2020** |
| Total population (billion) | 1.37 | 1.55 | 1.45 | 1.52 | 1.44 | 1.5 | 1.34 * | 1.41 * |
| Urbanization rate | 40% | 52% | 52% | 60% | 56% | 63% | 49% * | 61% * |
| GDP (trillion ¥) | 19 | 54 | 57 | 107 | 56 | 105 | 41 * | 101 * |
| Three-industry structure | 12:49:39 | 8:49:44 | | | | 4:46:50 | 9:47:44 * | 8:38:55 * |
| Effective irrigation area (million km$^2$) | 56.5 | 58.6 | 58 | 60 | 59.4 | 62 | 60.3 * | 69.2 * |
| Water requirements for Industry (billion m$^3$) | 149.8 | 191.1 | 165 | 170 | 160.5 | 171.8 | 144.7 ** | 103 ** |
| Water requirements for Agriculture (billion m$^3$) | 421.9 | 425.7 | 400 | 400 | 421.8 | 414.9 | 368.9 ** | 361.2 ** |
| Water requirements for Domestic (billion m$^3$) | 70.8 | 95.1 | 86 | 99 | 87.2 | 102.1 | 76.6 ** | 86.3 ** |
| Total water requirements (billion m$^3$) | 642.4 | 711.9 | 650 | 680 | 696.4 | 719.2 | 602.2 ** | 581.3 ** |

Notes: *: National Bureau of Statistics of China (https://data.stats.gov.cn/easyquery.htm?cn=C01, accessed on 31 December 2010). **: Ministry of Water Resources of the People's Republic of China (http://szy.mwr.gov.cn/gbsj/index.html, accessed on 31 December 2020).

## 2. Research Methods of Water Demand Forecasting

### 2.1. Research Status

Overall, domestic and international water resource forecasting approaches may be divided into two categories: classic mathematical models and comprehensive forecasting models [7,17–26].

The typical mathematical simulation technique involves developing a forecast model through numerical analysis of historical water use data. Scotland carried out planning work three times in the 1970s based on actual water use statistics to anticipate water demand in three stages, and the conclusions were very close to the actual situation [7]. In 2003, Joseph [17] developed the WaterGAP2 model to anticipate the water demands of the household, agricultural, and industrial sectors. A Sigmoid curve represents the home water process, and a hyperbola represents the industrial water process [17]. In 2010, in order to study the existing water demand in Xinzheng City and the state of urban water use, Wang et al. chose three economic constructions and development strategies [20], and then came up with a set of workable water use plans. In 2016, Thomas et al. utilized a linear gray transfer function to accurately forecast the average water demand of a single water-using family in the United States based on regional weather conditions, water pricing, and area economic development [19]. In 2018, the firefly algorithm was used by Wang et al. to anticipate water demand and successfully address a number of real-world engineering issues [21]. In 2019, Cheng et al. used the quadratic smoothing approach in a time series model based on current domestic water consumption data from 2005 to 2016 to forecast future demand for China's water resources [19]. This particular approach, however, is

based on a single equation model, which has a lively prediction time and is more prone to simulation than prediction.

The integrated and unified analysis of multiple socioeconomic and environmental variables has achieved a reality with the expansion of technology and the diversification of water demand, and comprehensive prediction models have gradually been created and developed [22–26]. The comprehensive forecast model gives a comprehensive forecast based on mathematical simulation forecasting between several scales or departments, along with specific parameter cooperation. In 2009, Zhang et al. combined projection pursuit technology with high-dimensional nonlinear and non-normal issues in water demand forecasting to construct a water demand forecasting model and employed immune evolutionary algorithms to choose key parameters in the model, which had broad applicability [23]. In 2017, Rathnayaka et al. increased the model's accuracy by incorporating the dynamics of urban residential water demand and its underlying variables into a model that predicts ultimate water demand at different scales [24]. In 2019, Li et al. developed a dynamic model of the urban water cycle system, investigated the interaction of the complete water cycle in the socioeconomic-ecological system, and anticipated future urban water demand [25]. In 2020, Marta et al. employed the Quadratic Almost Ideal Demand System (QUAIDS) to anticipate domestic water use in Spain in the belief that the QUAIDS model has a better fit and produces better forecast results than the regularly used linear, log-linear, and double-log water demand models [26].

### 2.2. Urgent Problems and Solutions

To summarize, although some research findings have been acquired in specific sectors, water resource prediction and analysis involve numerous disciplines and cover a large variety of aspects. China has not yet formed a mature discipline as a whole. The theoretical system of demand analysis and methodological study still require further development. This study organizes and summarizes historical domestic and foreign water resources literature [17–26], and then conducts a careful analysis of water resource demand projection. It concludes that the current issues mostly include:

- The divergence of the anticipated value is substantial, and one of the major reasons is the deviation of basic data. Because there is a certain divergence between the historical data and the actual value, there is an error between the forecast result and the real value.
- A lack of systematic and continuous prediction research leads to a long cycle (more than 30 years) covering three fields (national, regional, and watershed) and three levels (total amount, four departments, and sub-departments).
- Inadequate theoretical and technical support, insufficient summarization of consumption laws, and a hazy driving mechanism for the adjustment and optimization of the economic and social development model.
- There is a lack of understanding of supply-side constraints such as water resource endowment and carrying capacity, water conservation projects, management and macro-control, and other factors.

Exploring the quantitative relationship between these factors and water demand from the social, economic, and policy aspects affecting water resources, and establishing a medium and long-term forecasting system for water resources demand, is an important task for future water resources demand forecasting; it is also the main problem that this paper aims to solve.

## 3. Data Sources

Water resource forecasting requires a wide range of data from three primary categories: socioeconomic, national land, and water resources [11–16,27]. In terms of water resources data, data from the Food and Agriculture Organization of the United Nations (FAO), the European Union Statistics Office (EUROSTAT), the United Nations Statistics Division (UNSD), the Academy of Water Sciences, the Ministry of Water Resources, and the United

States Geological Survey are referred to in addition to national statistical bureaus. Data are obtained from the US Geological Survey, Japan's Ministry of Land, Infrastructure, Transport, Tourism, and others. The Food and Agriculture Administration, the National Bureau of Statistics, and the China Environmental Statistical Yearbook are the primary sources of agricultural statistics. Agricultural data are mostly derived from the International Food and Agriculture Administration, the China Statistics Bureau, and the China Environment Statistics Yearbook. Population data are based on the most recent national population data from the United Nations Census Bureau, the World Bank, the United Nations World Resources Institute, and the Central Intelligence Agency, which were all available online in 2018. Social and economic data, including global GDP, urbanization rate, industrial structure, arable land area, industrial and agricultural output, and other social wealth accumulation data, are primarily derived from data published online by national statistical offices, the World Bank, the Conference Board, and the Central Intelligence Agency. The Ministry of Land and Resources and the National Bureau of Statistics of the People's Republic of China provide data on land and resources.

## 4. The Modeling Framework

Given the existing difficulties in Section 2.2, the model technique used in the first phase of this study was to employ typical national data samples as "restricted data"; the second step was to develop a mathematical prediction model using gray systems, regression equations, and so on. Finally, a three-in-one water resource demand forecast model of "data constraint + mathematical prediction + actual revision" was created by combining scenario analysis, quota method, analogy method, and reference economic and social data to carry out "the actual revision".

### 4.1. Build Big Data on Water Consumption

The research team created a sizable water use data set to guarantee that the model's predictions were accurate. The main goals were long-scale water resources data, consumption data (water demand, water consumption, water use structure, etc.), economic and social data, and ecological and environmental data collected from the world's typical water resource countries, as well as economic and agricultural data in collaboration with relevant authorities. The huge data of water resources with a time scale of 70 years were finally created after screening, cleaning, and any necessary interpolation processing. The creation of the database provides a strong platform for the investigation of the relationship between the indicators and consumption laws.

### 4.2. Reveal the General Law of Water Resources Consumption

Based on long-scale and diverse data, and by analyzing the correlation between multiple factors and indicators that have affected water resource consumption in typical countries for decades, this paper summarizes the characteristics of water resource consumption in different development stages (generally represented by per capita GDP or urbanization rate), determines the key inflection point (singularity needs to be removed), and then prompts the general law of water resource consumption. The original "s" type approach of the Chinese Academy of Geological Sciences' global mineral resources strategic research center has been widely employed in medium and long-term demand forecasting in the field of energy and bulk mineral resources and has been proven to be highly accurate [21]. This study follows its principle and method.

### 4.3. Consolidate Key Indicators for Future Economic and Social Development

As determined in this study, the reliability of the four key indicators (independent variables) of economic development, population growth, scientific and technical progress, and consumer behavior has a direct impact on the accuracy of demand forecasting. Short time scales, complex sources, and substantial variances in data from multiple institutions are common issues with such measures. Forecasters must have a solid knowledge foundation

and extensive expertise, be able to present their own data selection and adjustment ideas, and be discussed and recognized by specialists in relevant subjects. Figure 1 depicts the water resource demand prediction processes and technological route framework.

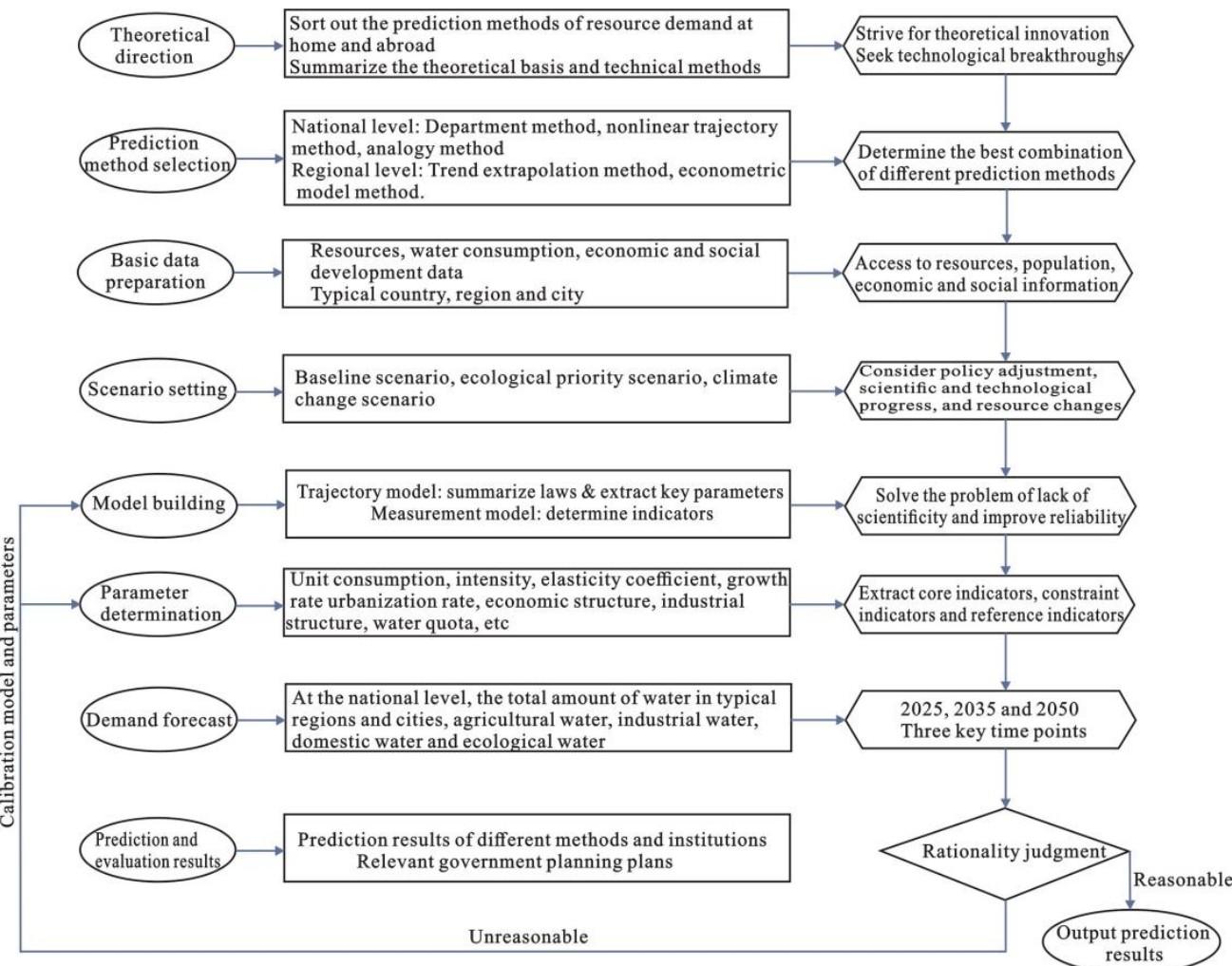

**Figure 1.** Technical route of water resource demand prediction.

## 5. Research on the Law of the Coupling Relationship between Water Resources, Economy, and Society

In order to analyze the coupling relationship between water resources, economy, and society according to agricultural water, industrial water, and domestic water (ecological water is not counted abroad, and this study is not conducted on ecological water), 15 countries, including China, the United States, Britain, France, Spain, Italy, Japan, Denmark, Poland, South Korea, Romania, India, Greece, South Africa, and Egypt, are selected as typical countries.

### 5.1. The Trend of per Capita Water Consumption and Consumption Intensity

According to an analysis of the link between per capita GDP and per capita water consumption in typical countries, there is a peak in per capita water consumption. Even though peak times vary by country, most of them (Figure 2) are around $10,000 per capita GDP (GAIKAI $, same below), and the equivalent per capita water resource use is 500–1000 m$^3$. The per capita water resource use has reached its pinnacle with economic expansion and is now close to 400 m$^3$.

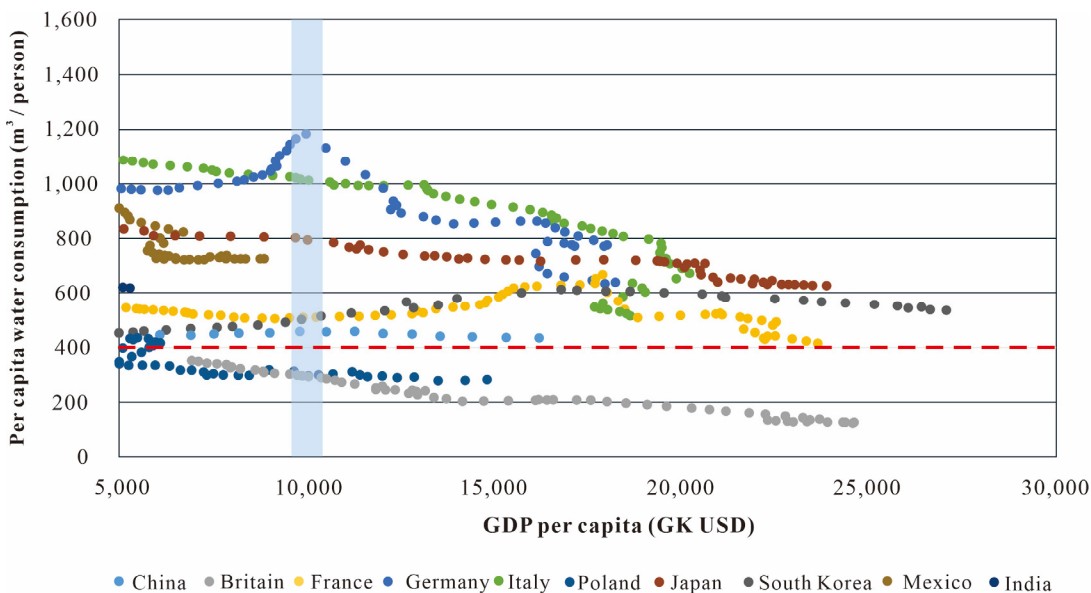

**Figure 2.** Relationship between per capita water consumption and per capita GDP.

With the growth of the economy and society, the consumption intensity of water resources (measured as water consumption per unit of GDP) exhibits a change law in the shape of an "L," similar to that of energy [28,29]. The "sharp drop point" of water resource consumption intensity (in the early stages of industrialization) is $5000 per capita GDP, and China's water resource consumption intensity is 1000–2000 m³/$10,000. The per capita GDP of $10,000 represents the "turning point" (mid-industrialization), the consumption intensity of water resources is 500–1000 m³/$10,000, the per capita GDP of $20,000 represents the "zero growth point" (late industrialization/Post Industrialization) (Figure 3), and the consumption intensity of water resources is 100–500 m³/$10,000, implying that the efficiency of water resource utilization is gradually declining. The regular changes mentioned above are related to the basic consumption characteristics of large unit consumption of agricultural water, easy control of industrial water efficiency, and small unit consumption of water in the service industry.

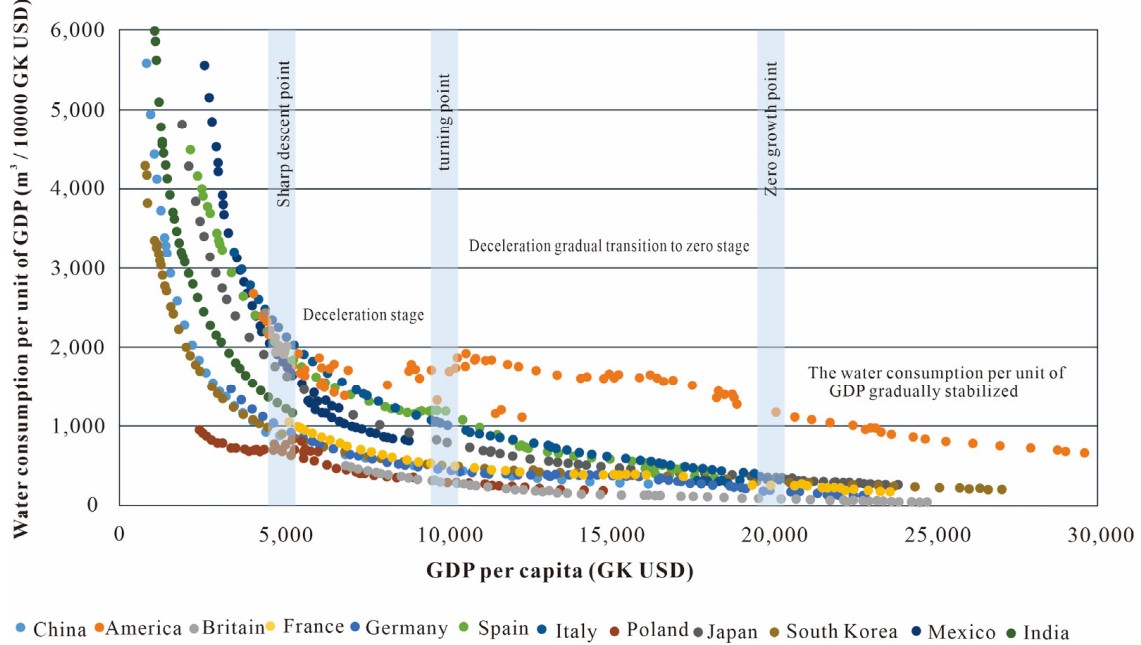

**Figure 3.** Relationship between water consumption intensity and per capita GDP.

*5.2. The Order of Peak Consumption*

- Agricultural water use

With 3000–4000 m$^3$ of water per hectare, the highest value of agricultural water demand per unit of cultivated land reflects the economic stage of US $10,000–12,000 per capita GDP. Later, as economic development, agricultural, and irrigation technology advanced, the amount of water used per unit of cultivated area in the majority of countries gradually decreased. It should be emphasized that each country's planting structure is unique, determined by precipitation, and the amount of water used per square meter of farmed land varies considerably, making it difficult to compare directly.

- Industrial water

With minimal variation between nations, the peak level of per capita industrial water consumption is roughly 200 m$^3$ and corresponds to the stage of development where the per capita GDP is between 15,000 and 17,000 $. Industrial water use increased quickly in the early stages of industrialization, and there was an inverted U-shaped relationship between per capita industrial water use and per capita GDP, meaning that as per capita GDP increased, industrial water use per capita first rose and then declined, indicating that as the economy advanced to a certain point, technological advancement encouraged the improvement of the industry.

- Domestic water

Peak domestic water consumption per person is correlated with a per capita GDP of $20,000–22,000 and an urbanization rate of 70–80%, or roughly 100–200 m$^3$, with significant regional variations. The rate of urbanization is a significant predictor of home water use. The rate of urbanization is rising, the urban population is growing, water consumption patterns are changing, and more people are using the public water supply, which has increased per capita home water use. The per capita household water consumption will start to slowly decrease as the pace of urbanization reaches a certain point, thanks to increased water use efficiency and modified water usage rules.

## 6. Analysis of China's Water Resource Demand in the Next 30 Years

*6.1. The Total Water Demand Shows a Downward Trend, and Domestic and Ecological Water Use Continues to Grow*

Building a correlation model, creating a non-trajectory approach, and projecting China's water resource demand over the next 30 years using the quota method, grey model, and other techniques are the technical steps taken in accordance with the technical route depicted in Figure 1. The major fundamental parameters are established based on the findings of numerous domestic and international institutions doing research on the future developments and trends of China's significant economic and social indicators. For information, see Table 2.

**Table 2.** Economic and social development indicators of China from 2020 to 2050.

| Items | 2020 * | 2025 | 2030 | 2035 | 2040 | 2045 | 2050 |
|---|---|---|---|---|---|---|---|
| GDP growth rate/% | 2.2 | 5 | 4 | 3 | 2 | 2 | 2 |
| GDP/trillion | 23.36 | 30.37 | 36.95 | 42.83 | 47.29 | 52.21 | 57.64 |
| Per capita GDP | 16,624 | 21,331 | 25,839 | 30,021 | 33,421 | 37,409 | 42,095 |
| Urbanization rate/% | 63.9 | 66.5 | 70.6 | 73.9 | 76.4 | 78.3 | 80 |

Note: total GDP and per capita GDP are in $; population and urbanization rate from the United Nations. *—actual data.

The forecast of the water resource demand in China for the next 30 years is based on the technological route and method system discussed above. The findings indicate that China's water use has reached a peak and will continue to decrease going forward. China's water consumption decreased significantly to 581.3 billion m$^3$ in 2020 as a result of the epidemic, with agricultural water, industrial water, domestic water, and ecological water

accounting for 62%, 18%, 15%, and 5% of that total, respectively, with 361.3 billion m$^3$, 103 billion m$^3$, 86.3 billion m$^3$, and 30.7 billion m$^3$. In 2035, it will be 534.9 billion m$^3$, of which 303.5 billion m$^3$ will be used for agricultural water, 85.4 billion m$^3$ for industrial water, 97.9 billion m$^3$ for domestic use, and 48.1 billion m$^3$ for ecological water, making up 57%, 16%, 18%, and 9% of the total. Agricultural water, industrial water, domestic water, and ecological water make up 281.1 billion m$^3$, 61.4 billion m$^3$, 101.4 billion m$^3$, and 60.8 billion m$^3$, respectively, or 56%, 12%, 20%, and 12% of the total volume in 2050. (Table 3 and Figure 4).

**Table 3.** Water resources consumption/demand in China from 1980 to 2050.

| Items 100 Million m$^3$ | 1980 * | 2000 * | 2010 * | 2020 * | 2025 | 2030 | 2035 | 2050 |
|---|---|---|---|---|---|---|---|---|
| Agricultural water | 3900 | 3784 | 3783 | 3613 | 3454 | 3191 | 3035 | 2811 |
| Industrial water | 500 | 1139 | 1447 | 1030 | 1100 | 957 | 854 | 614 |
| Domestic water | 100 | 575 | 672 | 863 | 911 | 955 | 979 | 1014 |
| Ecological water | - | - | 120 | 307 | 364 | 426 | 481 | 608 |
| Total | 4500 | 5498 | 6022 | 5813 | 5828 | 5529 | 5349 | 5047 |

Note: the agricultural water demand in the table is the result of normal water years; the statistical caliber of consumption, demand, and water consumption in this paper is the same. *—actual data.

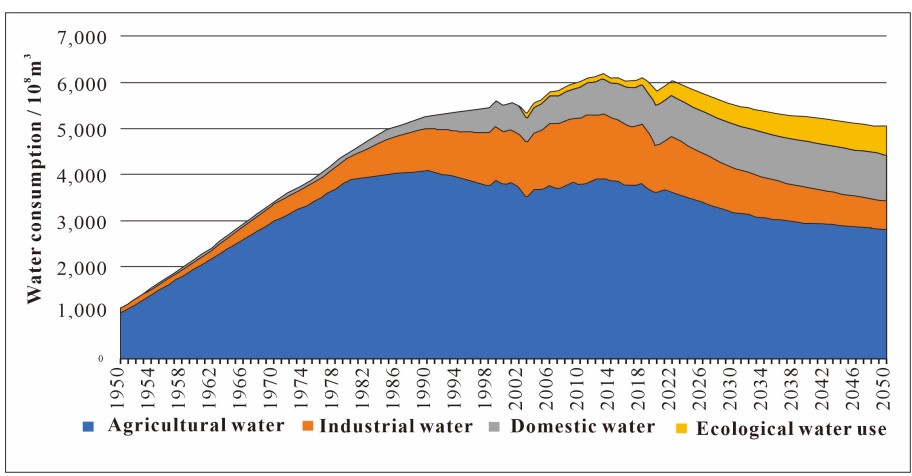

**Figure 4.** Consumption/demand of water resources and sector structure in China from 1950 to 2050.

As shown in Table 2, Figures 3 and 4, the forecast results are largely consistent with the limits to the growth of energy and mineral resources consumption predicted by Wang et al. [27,29], indicating that the forecast results are more in line with the law of China's economic and social development [6,9,17]. It also demonstrates that the law of China's economic and social development has the same demand for water as other foreign countries.

The prediction findings produced in this research are displayed in Table 3, and they are more accurate than the results of other studies' predictions [30–32]. For example, Jia et al. [30] utilized the trajectory approach to assess and estimate that China would essentially enter the "zero growth" stage of water demand in 2030. Yao et al.'s [31] pure mathematical models indicated that the water demand in 2030 will be 780 billion to 820 billion m$^3$. The sector forecasting method was utilized by Qian et al. to predict China's 711.9 billion m$^3$ water consumption in 2030 [32]. In comparison with the actual figures for 2016 (Table 1), Jia et al. received smaller data while Yao et al. and Qian et al. received larger data [30–32]. The reasons for the inaccurate prediction are as follows: firstly, the consumption law is not summarized based on the long period and large sample data, and the long-term change trend of water resource demand cannot be grasped; secondly, there is a deviation in the judgment of the future economic and social development (the basic indicator of prediction).

*6.2. The Intensity of Water Resources Development in North China Is Far beyond the Warning Line, and It Will Be More Severe in the Future*

The Ministry of Water Resources estimates that the overall amount of water resources in the country has averaged 2.8 trillion m$^3$ over many years, or around 5% of the globe's water. The amount of water used per person is approximately 2000 m$^3$, or one-fourth of the average amount worldwide. Overall, there is mild water scarcity.

The spatial distribution of water resources is greatly out of harmony with the spatial distribution of land resources, and there is a tremendous disparity between the two (economic and social activities). For instance, North China's land, population, and water consumption in 2020 were 16%, 12%, and 9% of the nation, respectively, yet its water resources represented only 3% of the total; Southwest China makes up 24% of the country's land area, 15% of its population, and 10% of its water use, yet it has up to 37% of the country's water resources (Figure 5).

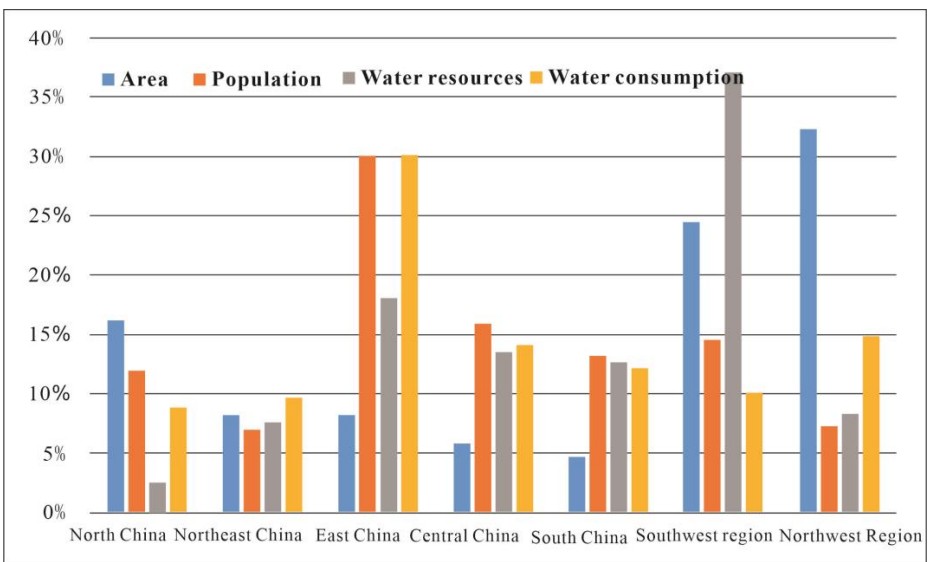

**Figure 5.** Proportion of geographical area, population, water resources, and water consumption in the country in 2020. North China: Beijing, Tianjin, Hebei, Shanxi, and Inner Mongolia; Northeast: Liaoning, Jilin, and Heilongjiang; East China: Shandong, Jiangsu, Anhui, Shanghai, Zhejiang, Jiangxi, and Fujian; Central China: Henan, Hubei, and Hunan; South China: Guangdong, Guangxi, and Hainan; Southwest: Chongqing, Sichuan, Yunnan, Guizhou, and Tibet; Northwest: Shaanxi, Gansu, Ningxia, Qinghai, and Xinjiang.

There are significant variances between geographical areas in terms of the pattern of water resource utilization. This study concentrates on Northwest China, which is the focus of the remarks, and North China, where the supply and demand condition of water resources is the most severe. In North China, agricultural water use will drop from 76% in 2000 to 58% in 2020 and then to 44% in 2050. Ecological water use, however, will rise sharply from 17% in 2020 to 35% in 2050 and will even surpass the combined use of industrial and domestic water (Table 4). In Northwest China, animal husbandry is advanced, and agricultural water use is still high, accounting for 89% and 83% of total water use in 2000 and 2020, respectively, and falling to 70% by 2050, but remains higher than the national average levels of 62% at present and 56% in 2050.

Due to the region's relatively underdeveloped economy and policy restrictions, industrial and domestic water use in Northwest China are significantly lower than the national average, at 4% and 6%, respectively, in 2020 and 2% and 12%, respectively, in 2050, which is still at a very low level. Based on specialized research on the future climate change and the changing trend of water resources in the region, the change in ecological water use in the region is predicted (Table 5).

**Table 4.** Water resources utilization structure in North China.

| Unit: m$^3$ | 2000 * | | 2020 * | | 2035 | | 2050 | |
|---|---|---|---|---|---|---|---|---|
| Water Sector | Consumption | Proportion | Consumption | Proportion | Consumption | Proportion | Consumption | Proportion |
| Agricultural water | 381 | 76% | 302 | 58% | 260 | 48% | 245 | 44% |
| Industrial water | 65 | 13% | 52 | 10% | 34 | 6% | 22 | 4% |
| Domestic water | 58 | 12% | 77 | 15% | 95 | 18% | 99 | 18% |
| Ecological water | - | - | 88 | 17% | 151 | 28% | 195 | 35% |
| Total | 504 | 100% | 518 | 100% | 540 | 100% | 562 | 100% |

Note: *—actual data.

**Table 5.** Utilization structure of water resources in Northwest China.

| Unit: m$^3$ | 2000 * | | 2020 * | | 2035 | | 2050 | |
|---|---|---|---|---|---|---|---|---|
| Water Sector | Consumption | Proportion | Consumption | Proportion | Consumption | Proportion | Consumption | Proportion |
| Agricultural water | 708 | 89% | 712 | 82% | 586 | 75% | 537 | 70% |
| Industrial water | 50 | 6% | 34 | 4% | 23 | 3% | 16 | 2% |
| Domestic water | 38 | 5% | 52 | 6% | 77 | 10% | 89 | 12% |
| Ecological water | - | - | 67 | 8% | 92 | 12% | 121 | 16% |
| Total | 796 | 100% | 865 | 100% | 778 | 100% | 763 | 100% |

Note: *—actual data.

As the static reference value for the calculation of water resources' development intensity (water consumption/water resources) in the next 30 years, the relatively optimistic national water resources volume of 3.16 trillion m$^3$ in 2020 (13% higher than the annual average) is used. The future changing trend of water resources' development intensity in each geographical region is analyzed. According to the findings, North China is the only geographical area that will continue to grow, going from 64% today to 70% in 2050. Other regions continued to fall below the internationally acknowledged ecological warning line of 40% of water resources development, reaching an average of less than 30% in 2050. (Figure 6).

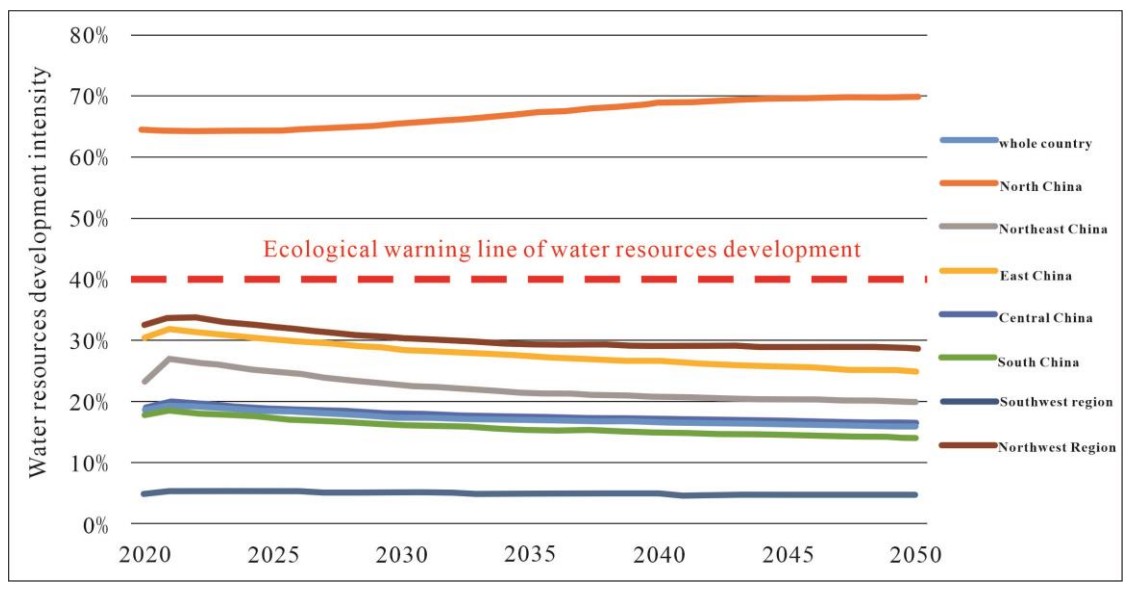

**Figure 6.** Trend of water resources' development intensity in each geographical division from 2020 to 2050.

## 7. Conclusions

(1) Based on energy prediction methods, we proposed the "three in one" production model, namely: "data constraint + mathematical prediction + actuarial prediction." The model integrates a wide range of disciplines, including economics, ecology, engineering, and mathematics. We anticipated water consumption in practice. Compared with the actual data (Tables 1 and 3), the strategy proved to be both unique and practical, which not only follows the economic development trajectory but also takes into account changes in the basic data.

(2) Peak demand for water resources in China (618.3 billion $m^3$) was reached in 2013, with demand varying substantially by sector. For example, agricultural water has traditionally been the primary source of water consumption in China, peaking in 1990, and will continue to account for more than half of the total demand by 2050. Industrial water use efficiency has already surpassed the world's advanced level; the trend shows a moderate fall, which will continue for a long time, and be less than domestic water by 2030. Domestic water use is gradually increasing in tandem with urbanization and growing living standards and is predicted to peak in 2030.

(3) According to projections, in 2030, 56% of the water will be consumed by agriculture, 12% by industry, 20% by life, and 12% by ecology, with the composition of water consumption in different sectors altering as society develops. It is suggested to further demonstrate the viability of decreasing agricultural water consumption and increasing ecological water consumption in this region. Furthermore, inter-provincial water transfer and optimal adjustment of water use by departments still need to focus on North China.

(4) In the future, North China is the only geographical area that will continue to grow, going from 64% today to 70% in 2050. Other regions will continue to fall below the internationally acknowledged ecological warning line of 40% of water resources development, reaching an average of less than 30% in 2050.

This study is the first attempt to use the forecasting method of energy resource demand in the field of water resources, hence it obviously has flaws that need to be fixed in future research.

**Author Contributions:** Conceptualization and project administration, Q.Y.; methodology, Q.Y. and Q.W.; software and data curation, Y.Z.; writing—original draft preparation, Q.Y. and Q.W.; writing—review and editing, Q.W. All authors have read and agreed to the published version of the manuscript.

**Funding:** This study is funded by Geology and Mineral Resources Survey Project: Research on Global Mineral Resources Strategy (DD20221795), General Project of Henan Natural Science Foundation (202300410278) and Ecological Configuration and Global Strategy of China Water Resources (DD20190652), and National Natural Science Foundation of China (71991485, and 71991480).

**Acknowledgments:** Xiaoqian Guo, Cailian Hao, Li Jiang, Wanli Xing, and Chao Liu from the Institute of Mineral Resources, Chinese Academy of Geological Sciences, have done a lot of basic support work. We would like to offer our thanks in the meantime.

**Conflicts of Interest:** The authors declare no conflict of interest.

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
