# Peer review of "Analysis of Demand and Structural Changes in China’s Water Resources over the Next 30 Years"

_water, doi:10.3390/w15040745_

Round 1

Reviewer 1 Report

Please clarify the nolvety and significance of this study. moreover, the advantage and disadvantage of your proposed mathematical prediction model and your created a water resources demand forecasting model. Thank you.

Author Response

We greatly appreciate your constructive comments and we have prepared a detailed reply to your comments. Specific amendments are listed in the Annex.

Reviewer 2 Report

1. As "In this paper, by introducing the energy demand forecasting method for the first time", is "energy demand" suitable for water forecasting as water and energy are two different resources.

2. "We collected long-term, multi-sample data from more than 20 countries", what are names of these countries,why  do you select these countries as the basis of China's water forecast.

3. As many regions have different development features, is the same method suitable for all regions (north China, northeast China, ...)

Author Response

(The authors gave the same response as above.)

Reviewer 3 Report

The article is interesting considering the subject and the data interpretation. Some comment refers to:

- Introduction can be extended by using new references interpretation and/or comments from point 2.

- The aims of the study is not clear presented.

- The article can be better systematized in maximum 4 points and conclusions.

- Please check the spelling mistakes (e.g. m3)

Author Response

我们非常感谢您的建设性意见,我们已准备好对您的意见进行详细答复。具体修正见附件。

Reviewer 4 Report

The abstract should be more concise and highlight to a greater extent the contribution of the article. A certain methodology and data are used to predict future water demand, but how can this analysis or the methodology used be useful? The abstract does not answer this question and it is difficult to identify the contribution of the article.

The introduction is very brief (in MDPI's article structure the literature review is part of the introduction) in which I consider that 2 important issues are missing:

·         Provide some data about the imbalance between water supply and demand, whose existence is indicated by the references. If possible, it would also be useful to explain what explains the water demand (agriculture, industry, urban consumption, etc.) that justifies this analysis.

·         Further development of the purpose of this article. It is justified that future water demand should be estimated and long-term planning undertaken, but the end of the introduction does not sufficiently explain how this prediction can be useful.

The data section should include more information about the variables used. It is not necessary to include all of them, but it would be appropriate to mention the most important ones. Otherwise, the section focuses on indicating the institution from which the data are obtained without providing almost no information about the data..

The results are shown in two different sections and are limited to describing them. The conclusions do not include anything that could be considered a discussion section, so that it is not possible to identify the complete contribution of the article. It is necessary to include a discussion section that highlights how the work carried out represents progress in comparison with the existing literature and in which the following questions are answered:

·         Is the methodology novel? If it has not been used in any other case, there is a methodological contribution. However, for this contribution to be significant, this methodology must have some advantage over the other techniques. A comparison with these techniques could be included in this section to highlight the value of the one used in the article.

·         Is the result obtained relevant to improve management? The introduction of the article explains that the current estimates of China's future water demand are outdated and that a new one is needed. Therefore, the analysis is justified, but it is necessary to compare the estimate obtained in the article with others and check that the results improve the information currently in use.

These are the two ways in which the discussion section would position this article in the literature, either through a methodological contribution, through the improvement of existing water demand estimates, or through a combination of both contributions.

Finally, the conclusions also focus on describing the results obtained. This section should briefly highlight the value of the methodology used and the usefulness of the results, which are the two key aspects of the article. In other words, the conclusions should be a brief section indicating that with the available data, there is a methodology that makes it possible to estimate future water demand accurately, thus allowing for improved current and future planning.

Author Response

(The authors gave the same response as above.)

Round 2

Reviewer 2 Report

the author has revised according to previous comments and the paper can be published.

Author Response

We appreciate your advice on how to make the paper better. We have looked through the entire text and made the necessary revisions.

Reviewer 4 Report

Thank you very much for the modifications made. However, there is still one major point for improvement. It is a simple but important issue. It is the methodological comparison with other existing forecasts not only from China, but also from other countries. The absence of such a comparison in the article makes it impossible for the reader to distinguish exactly what the article's contribution is. If the methodology is more effective than those previously used in other countries, this article has value in improving the estimation of water demand in more countries, not just China. Some review of the state of the art appears in section 2.1, but sections 5 and 6 should, at some point, establish that this methodology is effective and improves on those that have been used so far. The part on demand prediction is very good, but are the results obtained more accurate than they would have been with other existing methodologies? Sections 5 and 6 should answer this question and the conclusions should include why the proposed methodology is an improvement not only in China, but in general. It is complicated to make a comparison with the prediction of a different country that has applied another methodology, as two types of differences are combined, but it is necessary to clearly position the methodological proposal.

Author Response

(The authors gave the same response as above.)
